# Novel Insights into the Crosstalk between Mineralocorticoid Receptor and G Protein-Coupled Receptors in Heart Adverse Remodeling and Disease

**DOI:** 10.3390/ijms19123764

**Published:** 2018-11-27

**Authors:** Barbara M. Parker, Shelby L. Wertz, Celina M. Pollard, Victoria L. Desimine, Jennifer Maning, Katie A. McCrink, Anastasios Lymperopoulos

**Affiliations:** 1Laboratory for the Study of Neurohormonal Control of the Circulation, Department of Pharmaceutical Sciences (Pharmacology), College of Pharmacy, Nova Southeastern University, Fort Lauderdale, FL 33328, USA; abs255@yahoo.com (B.M.P.); sw1541@yahoo.com (S.L.W.); cp1743@yahoo.com (C.M.P.); vdesimine@yahoo.com (V.L.D.); jm3706@mynsu.nova.edu (J.M.); jm3706@yahoo.com (K.A.M.); 2Present address: Jackson Memorial Hospital, Miami, FL 33136, USA; 3Present address: Massachusetts General Hospital, Boston, MA 02114, USA

**Keywords:** adverse remodeling, aldosterone, cardiac myocyte, crosstalk, G protein-coupled receptor (GPCR), GPCR-kinase (GRK), heart failure, inflammation, mineralocorticoid receptor, myocardial infarction, oxidative stress, signal transduction

## Abstract

The mineralocorticoid hormone aldosterone regulates sodium and potassium homeostasis but also adversely modulates the maladaptive process of cardiac adverse remodeling post-myocardial infarction. Through activation of its mineralocorticoid receptor (MR), a classic steroid hormone receptor/transcription factor, aldosterone promotes inflammation and fibrosis of the heart, the vasculature, and the kidneys. This is why MR antagonists reduce morbidity and mortality of heart disease patients and are part of the mainstay pharmacotherapy of advanced human heart failure. A plethora of animal studies using cell type–specific targeting of the MR gene have established the importance of MR signaling and function in cardiac myocytes, vascular endothelial and smooth muscle cells, renal cells, and macrophages. In terms of its signaling properties, the MR is distinct from nuclear receptors in that it has, in reality, two physiological hormonal agonists: not only aldosterone but also cortisol. In fact, in several tissues, including in the myocardium, cortisol is the primary hormone activating the MR. There is a considerable amount of evidence indicating that the effects of the MR in each tissue expressing it depend on tissue- and ligand-specific engagement of molecular co-regulators that either activate or suppress its transcriptional activity. Identification of these co-regulators for every ligand that interacts with the MR in the heart (and in other tissues) is of utmost importance therapeutically, since it can not only help elucidate fully the pathophysiological ramifications of the cardiac MR’s actions, but also help design and develop novel better MR antagonist drugs for heart disease therapy. Among the various proteins the MR interacts with are molecules involved in cardiac G protein-coupled receptor (GPCR) signaling. This results in a significant amount of crosstalk between GPCRs and the MR, which can affect the latter’s activity dramatically in the heart and in other cardiovascular tissues. This review summarizes the current experimental evidence for this GPCR-MR crosstalk in the heart and discusses its pathophysiological implications for cardiac adverse remodeling as well as for heart disease therapy. Novel findings revealing non-conventional roles of GPCR signaling molecules, specifically of GPCR-kinase (GRK)-5, in cardiac MR regulation are also highlighted.

## 1. Introduction

Aldosterone exerts important effects in various organ systems outside the kidneys, its primary target organ [1]. Among these systems is the cardiovascular system, of which both the heart and the vasculature are direct targets of aldosterone’s actions [2]. All of the genomic effects of aldosterone are mediated by the mineralocorticoid receptor (MR), resulting in altered gene expression that affects vascular tone/blood pressure, cardiac contractility, and ventricular wall remodeling [3]. Specifically, in the cardiovascular system, the MR is expressed in vascular endothelial and smooth muscle cells of murine kidneys and in the human heart, including cardiomyocytes, coronary endothelial and vascular smooth muscle cells, fibroblasts, and immune cells (e.g., macrophages, monocytes, etc.) [4,5]. Of note, cortisol, whose plasma levels are normally hundreds of time higher than aldosterone’s, binds to, and activates the MR with similar affinity to that of aldosterone [6,7]. Thus, MR hyperactivity is prevented by the presence and activity of 11β-hydroxysteroid dehydrogenase type 2 (11-βHSD2), which converts cortisol to the MR-inactive cortisone [7]. Cardiac myocytes appear to express very little 11-βHSD2, which means that the cardiomyocyte-residing MR may primarily be stimulated by cortisol rather than aldosterone [8]. Nevertheless, direct effects of aldosterone in cardiac myocytes have been documented and the MR plays important roles in cardiac physiology [9,10]. Since its effects are genomic, MR gene expression effects take at least several hours to manifest, but aldosterone is known to exert also more rapid, transient, non-genomic effects via other receptors, including GPER (G protein-coupled estrogen receptor) [11,12,13,14]. Contrary to its clearly defined function in the kidneys promoting sodium (and water) reabsorption and potassium excretion, the function of the MR in the normal healthy heart is poorly understood [13,15]. It has been shown to regulate cardiomyocyte growth and cardiac electrical conduction [5,6]. Nevertheless, a large body of evidence, both from transgenic mouse models of chronic pressure overload or myocardial infarction (MI) with manipulated MR expression levels and from large scale clinical trials of MR antagonists (MRAs), clearly documents the role of the MR in cardiac pathophysiology. The present review provides an overview of the role and signaling of the MR in cardiac pathophysiology with a particular emphasis on adverse remodeling. It also discusses the experimental evidence for MR’s cross-talk with cardiac G protein-coupled receptors (GPCRs), highlighting novel, cardiac-specific aspects of MR signaling that can be exploited for cardiovascular disease therapy.

## 2. MR in Cardiac Adverse Remodeling

Aldosterone directly induces hypertrophy, ventricular remodeling, arrhythmias, and ischemia in the myocardium. Importantly, these effects are mostly independent of aldosterone’s systemic hemodynamic effects [5,16]. Together with high salt (sodium), aldosterone increases myocardial inflammation via upregulation of the pro-inflammatory cytokines tumor necrosis factor (TNF)-α, interleukin-1β, and transforming growth factor (TGF)-β [17,18]. These effects are in part mediated by serum- and glucocorticoid-induced protein kinase (SGK)-1 and transcription factors nuclear factor (NF)-κB and activator protein (AP)-1 [18,19]. Collagen and pro-fibrotic factor synthesis, including connective tissue growth factor, TGF-β, plasminogen activator inhibitor (PAI)-1, matrix metalloproteinase (MMP)-2, and TNFα, also increases upon aldosterone/salt administration in the rat myocardium [18,20,21,22]. In addition, oxidative stress, as evidenced by nicotinamide adenine dinucleotide phosphate (NADPH) oxidase (NOX) activity and reactive oxygen species (ROS) production, is increased, contributing to the cardiac inflammation and fibrosis induced by aldosterone [5]. Moreover, MR activation stimulates apoptosis and induces coronary vasoconstriction in animal hearts [23,24]. The underlying mechanism for the reduced coronary blood flow by the MR is presumed to be impaired endothelium-dependent, nitric oxide (NO)-mediated vasodilatation due to decreased NO production [23].

The high-salt requirement for most of the aforementioned effects of aldosterone is thought to stem from the fact that high sodium stimulates oxidative stress, which, in turn, activates the cardiac MR [25]. Crosstalk between the MR and the angiotensin II type 1 receptor (AT_1_R), a member of the GPCR superfamily, has also been implicated in aldosterone’s damaging effects in the heart [26]. Perhaps the most solid evidence for the pivotal role of the MR in heart disease comes from the remarkable cardiac benefits the MRAs, specifically spironolactone and eplerenone, have been demonstrated to exert clinically and in animal studies. MRAs prevent or significantly attenuate cardiac inflammation, fibrosis, and oxidative stress induced by aldosterone [1,5,6]. They also reduce the risk of cardiac arrhythmias in animal models of heart disease [5]. Notably, these beneficial effects of the MRAs appear independent of any effects on systemic hemodynamics, which strongly indicates that they are due to direct cardiac (or vascular) MR blockade [1,5,15,16].

The MR is present in cardiac myocytes and functions essentially as a high-affinity cortisol receptor, since 11-βHSD2 is significantly under-expressed in these cells [7]. However, aldosterone still plays an important role in regulation of cardiac output. Studies in cardiomyocyte-restricted MR-knockout mice showed that the absence of the MR led to improved cardiac healing, preventing adverse remodeling, cardiac hypertrophy, contractile dysfunction, and maladaptive gene expression post-myocardial infarction (MI) [27]. Cardiac inflammation and apoptosis were also reduced early after the MI in these mice, and were accompanied by improved left ventricular filling pressures, end diastolic and end systolic volumes, and ejection fraction [27]. Immediate pharmacological blockade of the MR also ameliorates cardiac healing post-MI by reducing cardiac inflammation [28] and genetic ablation of the cardiomyocyte MR protects the heart in the transaortic constriction (TAC) model of pressure overload [29]. Notably, in the latter animal model of heart failure, the absence of cardiomyocyte MR only improved cardiac function without affecting cardiac hypertrophy, fibrosis, apoptosis, or inflammation post-TAC [29]. Thus, the cardiomyocyte-residing MR seems to affect cardiac function, while cardiac MR expressed in other cardiac cell types (e.g., fibroblasts, endothelial cells, infiltrated immune cells) regulates cardiac adverse remodeling. Indeed, the cardiomyocyte MR is essential for the primary inflammatory response and recruitment of inflammatory cells to the heart associated with high salt-induced cardiac remodeling [30]. Studies in transgenic cardiomyocyte MR-overexpressing mice corroborate the findings in cardiomyocyte MR-knockout mice. Genome-wide analyses revealed that connective tissue growth factor (CTGF) and the neutrophil gelatinase-associated lipocalin are among the early cardiac remodeling-associated MR target genes upregulated by chronic aldosterone treatment (despite the preponderance of glucocorticoid receptors in the heart) [31,32]. Of note, the MR does not seem to affect normal cardiac development or function. Cardiomyocyte MR-knockout mice have normal systolic and diastolic functions and cardiac dimensions [1]. When challenged with high salt, however, the inotropic and chronotropic functions of the MR-knockout hearts are dysregulated [31], which is consistent with evidence in isolated cardiomyocytes for aldosterone-dependent increases in positive inotropy and chronotropy [33,34,35]. The effect of the MR on heart rate is also modulated by the glucocorticoid receptor and oxidative stress [33]. The two steroid receptors act synergistically to regulate T-type and L-type calcium channel expression and activity, thereby increasing risk of arrhythmias in the myocardium [33]. Cardiac-specific MR overexpression leads to a high rate of sudden cardiac death in mice via reduced potassium transient outward and increased L-type calcium currents resulting in prolonged repolarization (refractory period) [36]. In addition, cardiomyocyte-specific MR overexpression causes NOX-dependent, ROS-mediated coronary endothelial dysfunction [37]. Additionally, both the MR and 11-βHSD2 are upregulated in rats post-MI, and, in response to a high-salt diet, cardiac MR expression is elevated in heart failure and atrial fibrillation patients [6,38,39,40,41].

Finally, additional effects by the MR expressed in other cardiovascular tissues outside the heart, indirectly, but still significantly, contribute to cardiac adverse remodeling. For instance, the MR promotes endothelial dysfunction in high cholesterol diet-induced atherosclerosis in mice, in atherosclerotic monkeys, and in models of experimental thrombosis [42,43]. Chimeric low-density lipoprotein receptor (LDLR)-knockout mice with MR-knockout bone marrow cells have reduced atherogenesis both basally and in response to angiotensin II [44]. In humans, polymorphisms of the aldosterone synthase (CYP11β2) gene have been associated with atherosclerotic plaque size, and plasma aldosterone was the only independent predictor of plaque progression in one large study [45]. Another important cell type with substantial endogenous MR expression and contributing to cardiac adverse remodeling is the macrophages [46]. They also express glucocorticoid receptor but not 11-βHSD2. Thus, under normal circumstances, macrophage MR is stimulated by cortisol, similarly to the cardiomyocyte MR [47]. Deletion of macrophage MR changes the baseline expression of several pro-inflammatory genes, but, interestingly, does not affect macrophage recruitment/infiltration into the diseases myocardium, indicating that the MR operating in other cardiac cell types, e.g., endothelial cells, contributes to macrophage infiltration in deoxycorticosterone/high salt-treated hearts [48,49,50]. Of note, even the MR in T-lymphocytes has been implicated in aldosterone-induced Th17-mediated immune activation, which might be part of the overall MR-driven cardiac inflammation [51,52].

In summary, deletion or inactivation of the MR gene attenuates left ventricular dilatation, cardiac hypertrophy, and heart failure progression, whereas overexpression of the MR in cardiomyocyte-specific MR overexpression promotes cardiac adverse remodeling, heart failure progression, and development of arrhythmias [27,29,36].

## 3. GPCR Signaling and MR Function

The human MR is a 984-amino acid cytoplasmic protein with three functional domains: the N-terminal domain (NTD) that regulates transcriptional activity of the receptor, the DNA-binding domain (DBD) involved in the binding of the promoter of the target gene, and the ligand-binding domain (LBD) responsible for hormone binding [3]. In the nucleus, the MR depends on numerous molecular co-regulators to activate and regulate its target genes that carry the (shared with the glucocorticoid receptor) glucocorticoid response element (GRE) sequence in their promoters [53]. The MR also undergoes post-translational modifications, such as phosphorylation, SUMOylation, ubiquitination, etc., which also play important roles in regulation of its transcriptional activity and of its ligand binding specificity/affinity [54]. MR activity is also affected by factors other than its ligands, including protein kinase A (PKA), Rac-1, ubiquitin conjugating enzymes, and other factors involved in the regulation of diverse nuclear receptors [53,54,55,56]. Additionally, as mentioned above, high salt (sodium) concentrations lead to MR activation, even in the absence of any hormone/ligand [1,25], and result in cardiac fibrosis and inflammation.

One of the most powerful physiological stimuli for the synthesis and secretion of aldosterone, and the last step in the renin-angiotensin-aldosterone system axis, is angiotensin II activation of the AT_1_R, a G_q/11_ protein-coupled receptor [57,58,59,60,61]. Specifically, the AT_1_R promotes aldosterone production in the adrenal cortex through G_q/11_ protein, i.e., diacylglycerol (DAG) and inositol trisphosphate (IP_3_) signaling, but also through βarrestin1 signaling to extracellular signal-regulated kinases (ERK)-dependent steroidogenic acute regulatory (StAR) protein upregulation [62,63,64,65,66]. Therefore, there is considerable (indirect) crosstalk between the MR and GPCRs at the level of the former’s natural hormone ligand regulation. However, there is substantial evidence for direct regulation of GPCR signaling mediators by the MR, as well. Apart from MR interactions with the epidermal growth factor receptor (EGFR), a receptor tyrosine kinase (RTK), which are well characterized [67,68,69], GPR30 or GPER, an estrogen-responsive GPCR, serves as a membrane receptor for aldosterone [70,71]. G_q_ protein-coupled receptor signaling-activated protein kinase C (PKC)-α also binds aldosterone directly (i.e., in an MR-independent manner), which leads to its auto-phosphorylation [72]. Aldosterone is known to activate mitogen-activated protein kinases (MAPKs), which play significant parts in GPCR signaling in all tissues, including the heart. Thus, the extracellular signal-regulated kinases (ERK)1/2 are activated by aldosterone in various cell types and tissues, including in vascular smooth muscle cells [73] and in cardiac myocytes [74,75]. In the latter cells, this leads to hypertrophy [75,76]. p38 MAPK is another MAPK activated by aldosterone via the MR in vascular smooth muscle cells [77], where it leads to fibrosis through NADPH stimulation. In fact, p38 MAPK blockade counters the high salt diet-induced deleterious cardiovascular effects in spontaneously hypertensive rats [78]. Contrary to PKCα, which directly binds aldosterone, PKCδ and PKCε are activated by aldosterone via MR-induced EGFR transactivation [79]. Finally, protein kinase D (PKD)-1 activation leading to cardiac hypertrophy has also been linked to the MR-EGFR crosstalk in aldosterone-treated cardiac myocytes [80], whereas, in vascular endothelial cells, aldosterone enhances nitric oxide production via MR- and phosphoinositol 3-kinase (PI3K)-dependent endothelial nitric oxide synthase (eNOS) phosphorylation [81].

As mentioned above, the MR undergoes several stimulus-induced post-translational modifications, most frequently direct phosphorylation, which underlies several rapid signaling events induced by aldosterone. Phosphorylation of co-factors required for MR transcriptional activity also plays an important role. Regulation of these phosphorylation events by GPCRs and GPCR-activated signaling molecules provides the basis for the opposite direction of GPCR-MR crosstalk to the one discussed above, i.e., GPCR-dependent regulation of the MR. Indeed, protein kinase A (PKA), activated by G_s_ protein-coupled receptors and inhibited by G_i_ protein-coupled receptors, induces dissociation of heat shock protein (Hsp)-90 from the MR [82] (Figure 1). This event is normally required for MR translocation to the nucleus (Faresse). Furthermore, the steroid receptor co-activator (SRC) family—comprising SRC1, SRC2, and SRC3—is another group of proteins required for transcription by nuclear receptors, including the MR and PKA phosphorylates SRC2 resulting in its ubiquitination and subsequent degradation [83]. Even the ERKs, which can be activated by the aldosterone-induced MR crosstalk with the EGFR (see above), phosphorylate the MR itself, thereby modulating MR protein stability (proteasomal degradation) and closing a negative feedback MR regulatory loop [84] (Figure 1). Moreover, serine-843 located within the LBD of the MR gets phosphorylated by an unidentified kinase, preventing MR binding to, and activation by aldosterone in renal intercalated cells [85]. Upon volume depletion (hypovolemia), the AT_1_R decreases Ser843 phosphorylation of the MR via protein phosphatase (PP)-1 activation in these cells, in order to increase chloride reabsorption, inhibit potassium excretion, and ultimately restore (increase) plasma volume [85]. Of note, however, this Ser843 phosphorylation event is renal intercalated cell-specific, and purportedly absent in cardiac myocytes [85].

Agonist-activated GPCRs are phosphorylated by a family of serine/threonine kinases collectively known as GPCR-kinases (GRKs). This phosphorylation enhances the affinity of the receptor for binding to the adapter proteins β-arrestins, which sterically hinder G protein coupling and activation, thereby conferring receptor functional desensitization [86]. Several GRKs are now known to phosphorylate non-GPCR substrates (the so-called “non-canonical” GRK actions [87]). There are seven mammalian GRKs (GRK1-7), all of which share a common structural architecture with a well-conserved, central catalytic domain (≈270 aa), similar to that of other serine-threonine kinases, flanked by an amino-terminal (NT) domain (≈185 aa) and a variable length carboxyl-terminal (CT) domain (≈105–230 aa) that contains specific regulatory sites [88,89]. The conservation of length and specific amino acids in the NT domain suggests that this region is involved in specific receptor recognition and binding and in intracellular membrane anchoring. The CT domain of GRKs contributes to their subcellular localization and agonist-dependent translocation by favoring their interaction with lipids and other membrane proteins. GRK2, GRK3, and GRK5 are ubiquitously expressed, including in the heart where GRK2 and GRK5 represent the most abundant isoforms [90,91]. When inactive, GRK2 and GRK3 are in the cytoplasm and need to interact with the free Gβγ subunits of activated heterotrimeric G proteins in order to translocate to the cell membrane and phosphorylate agonist-occupied GPCRs [90,91,92]. In contrast, GRK5 forms direct ionic interactions with the cell membrane phospholipids thanks to a highly basic (lysine-rich) region of its molecule, such that it is anchored to the plasma membrane even when inactive [91]. The mechanism of its activation by GPCRs, specifically by the β_2_-adrenergic receptor (AR), was elucidated recently [93]. Of note, GRK5 is located also in the cell nucleus, thanks to a nuclear localization/DNA binding sequence (NLS) it contains, where it can affect gene transcription via epigenetic mechanisms [94,95,96].

In transfected renal cells, the human MR has been shown to increase β_2_AR-dependent intracellular cyclic adenosine monophosphate (cAMP) levels via G_sα_ protein upregulation and GRK3 downregulation [97]. In murine hearts in vivo, the MR has been documented to promote heart failure by activating GRK2-dependent cardiac apoptosis and GRK5 nuclear accumulation-dependent cardiac hypertrophy [76] (Figure 1). These non-canonical, deleterious GRK effects appear to be mediated by an MR-induced, c-Src kinase-dependent transactivation of the AT_1_R in the heart [76] (Figure 1). Importantly, the authors of that study correlated the peripheral lymphocyte GRK2 levels, known to reflect myocardial GRK2 levels, of heart failure patients with MRA (spironolactone) treatment and found that patients treated with spironolactone had significantly lower peripheral lymphocyte GRK2 levels compared to non-MRA treated patients [76]. This probably reflects the better cardiovascular status of heart failure patients conferred by the MRA treatment.

In addition to GRK2 and GRK5 modulation by the cardiac MR, very recent data from our laboratory indicate that the opposite, i.e., cardiac MR regulation by GRKs, can occur as well. Indeed, we have found that GRK5, but not GRK2, phosphorylates the MR in H9c2 rat cardiomyoblasts and in adult rat venrtricular myocytes, inhibiting its transcriptional activity [98] (Figure 1). Moreover, this non-canonical effect of GRK5 is enhanced by β_2_AR activation. In contrast, GRK2 phosphorylates and desensitizes the non-genomic aldosterone receptor GPER [98]. Importantly, GRK5 appears necessary for the protective effects of the MRAs (eplerenone) against aldosterone’s deleterious effects in cardiomyocytes (apoptosis, oxidative stress, etc.) [98]. Of note, the GRK5-mediated MR phosphorylation occurs in the cytoplasm and seems to interfere with the ability of the MR to translocate to the nucleus to activate gene transcription [99]. Thus, the MR functional blockade by GRK5 is topologically independent from the kinase’s own nuclear/genomic effects, which can be harmful (i.e., pro-hypertrophic) in the heart. Therefore, GRK5 can counter the deleterious effects of the MR, thereby augmenting the beneficial actions of the MRA’s in the heart. This is another line of evidence supporting a beneficial, rather than detrimental, role for GRK5 in the myocardium. Indeed, enhanced GRK5 activity has been associated with favorable outcomes, similar to those of beta-blockers, in human heart failure [100], and GRK5 also inhibits cardiac NFκB, thereby attenuating inflammation and hypertrophy in the heart [101,102,103,104]. In diametric contrast, every action of GRK2 in the myocardium uncovered so far appears deleterious for cardiac function or structure [89,90,91,92].

## 4. Therapeutic Implications of GPCR-MR Crosstalk for Heart Disease

From the preceding sections, it becomes evident that there is a considerable amount of GPCR crosstalk with the MR in the heart, which can have enormous pathophysiologic and, consequently, therapeutic (in the context of heart disease) implications. From the perspective of the MR regulating downstream GPCRs and GPCR signaling mediators/regulators (Figure 1), therapeutic targeting of proteins directly activated by aldosterone like GPER, EGFR, and PKC has the potential of combating several non-genomic actions of aldosterone, which are as deleterious for the myocardium as its classic genomic actions, e.g., EGFR transactivation-mediated fibrosis, PKC-mediated hypertrophy, etc. Targeting of several of these signaling mediators is already being pursued for heart failure therapy, independently of their molecular connections with the cardiac MR. However, the main drawback of targeting these molecules is that the equally (if not worse) cardiotoxic actions of the cardiac MR, activated by aldosterone, cortisol, or no specific ligand (e.g., oxidative stress, hyperkalemia), are left unopposed.

For this reason, it is imperative that inhibition of the cardiac MR downstream signaling targets be combined with blockade of the activity of the cardiac MR itself, i.e., an MRA. Given the significant extent of GPCR signaling crosstalk occurring also upstream of the cardiac MR (Figure 1), targeting of GPCR signaling mediators/regulators that affect cardiac MR activity might also have therapeutic potential, at least in that it might act synergistically or additively with an MRA. In that vein, PKA inhibition or GRK5 stimulation in cardiomyocytes may augment the beneficial effects of MR antagonists in heart disease, since PKA activates and GRK5 inhibits cardiac MR transcriptional activity (Figure 1). Indeed, GRK5 appears indispensable for eplerenone’s cardioprotective effects in ARVMs [98]. Of note, β_2_AR stimulation in the cardiomyocyte, which can be achieved with agents currently used in clinical practice (i.e., the anti-asthmatic β_2_AR-selective agonists), can potentially lead to MR blockade in the heart, via GRK5 activation (Figure 1). β_2_AR-activated GRK5 not only directly phosphorylates and inhibits the cardiac MR, but also (indirectly) suppresses PKA activity by desensitizing the β_2_AR (i.e., terminating the receptor’s G_s_ protein signaling that activates PKA). Not to mention, the cardiac β_2_AR is purportedly capable of switching its coupling from G_s_ to G_i_ proteins, an event that would also suppress PKA activity (due to inhibition of cAMP synthesis) [105].

## 5. Conclusions and Future Perspectives

Although more studies are certainly needed to further elucidate the molecular and signaling connectome of the cardiac MR, two things are known for sure. The first is that the cardiac MR exerts overall negative effects in the myocardium, in particular in the diseased or injured myocardium (e.g., post-MI); thus, all of its actions, direct and indirect, genomic and non-genomic, need to be blocked in heart disease. This is why MRA drugs have been and continue to be so successful in human advanced heart failure therapy. The other proven and well-documented fact about the cardiac MR is that it displays significant signaling crosstalk with GPCRs and GPCR signaling components, either being upstream of the latter (i.e., modulating them) or being downstream (i.e., its activity being under the control of the latter). At the same time, there is a need for novel and improved MRA agents to successfully treat heart failure in humans, since the currently-used agents are riddled with several adverse effects (e.g., hyperkalemia, renal complications, etc.). Indeed, novel non-steroidal MRAs (e.g., finerenone), which are purportedly more potent and specific inhibitors at the MR, are currently in development for human heart failure treatment. Given that the MR is expressed throughout the cardiovascular system with various effects in each individual tissue/cell type, it would be advantageous for the pharmaceutic industry to develop novel agents that specifically block the MR only in the myocardium. To this end, targeting the signaling crosstalk mechanisms between the MR and GPCRs that occur specifically in cardiac myocytes could be instrumental. With the realization that an MRA agent is not enough to fully counter the cardiotoxic actions of the MR or of aldosterone and as more data on the biochemistry and molecular (patho) physiology of the MR in the heart become available, the pharmaceutical industry’s odds of coming up with new and better MR-targeting drugs for heart disease therapy are looking pretty good.

## Figures and Tables

**Figure 1 ijms-19-03764-f001:**
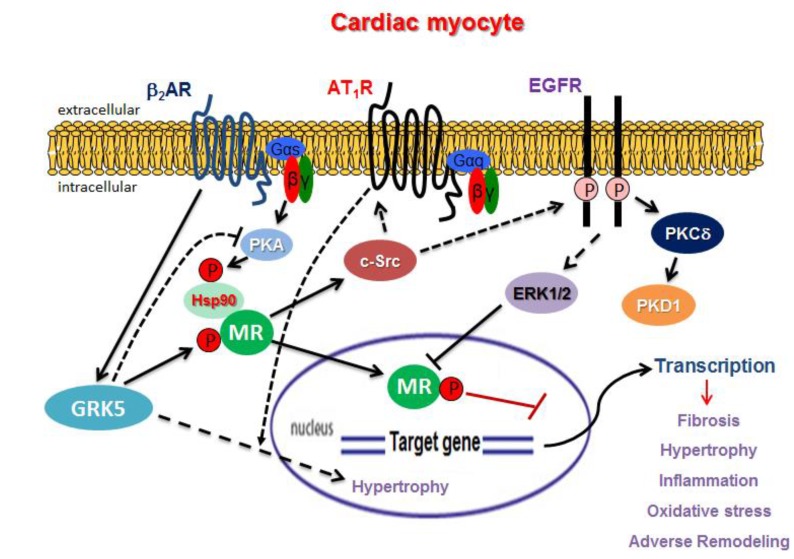
Important GPCR-related molecular pathways involved in crosstalk with the MR in cardiac myocytes. Aldo: Aldosterone; P: Phosphorylation. Solid arrows indicate direct effect, whereas dotted arrows indicate indirect (through additional intermediate proteins) effect. See text for details and for all other molecular acronym descriptions.

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
