# Peer review of "Novel Insights into the Crosstalk between Mineralocorticoid Receptor and G Protein-Coupled Receptors in Heart Adverse Remodeling and Disease"

_ijms, 2018, doi:10.3390/ijms19123764_

Reviewer 1 Report

Parker et al wrote a very interesting review on the crosstalk between MR and GPCRs in the heart, strenghtening also the therapuetic implications. I find the review very well written and of general interest. In my opinion it can be published as it is. 

I have only one question for the authors: are there any evidences of G-protein independent signaling events of GPCRs modulating MR in the heart (e.g. ß-arrestin crosstalk with MR)? 

Author Response

We thank this reviewer for their kind words about the quality of our work. Regarding his/her question, we are not aware of any studies on arrestin-mediated signaling by GPCRs to the MR in the heart. To our knowledge, the only evidence for beta-arrestin interactions with the MR comes from our own studies on beta-arrestin1 involvement in AT1R-dependent aldosterone production regulation in the adrenal cortex and from the study by Cannavo et al. (Ref. 76 of our manuscript), which showed that the MR transactivates the AT1R in neonatal rat ventricular myocytes through beta-arrestin1-dependent c-Src recruitment. This is discussed in lines 248-249 (page 6) of our manuscript.

We hope this response satisfies this reviewer.  

Reviewer 2 Report

In this manuscript titled, "Novel insights into the crosstalk between mineralocorticoid receptor and G protein-coupled receptors in heart adverse remodeling and disease ", Barbara M. Parker et al., authors summarize the current literature on mineralocorticoid receptor and G protein-coupled  receptors in heart adverse remodeling and disease. Authors analyzed the evidence of summarizes the current experimental evidence for this GPCR-MR crosstalk in the heart and discusses its pathophysiological implications for cardiac adverse remodeling as well as for heart disease therapy. This review is well written. For the study the presented data are quite sufficient.

1.         In part 2. MR in cardiac adverse remodeling. Authors should supply a diagrammatic sketch to summary the effect of aldosterone contribute to the cardiac inflammation and fibrosis.

2.         According to line106-108, authors should supply the data of Cardiac inflammation and apoptosis in different phase of MI mice, as well as left ventricular filling pressures, end diastolic and end systolic volumes, and ejection fraction.

3.         In part 5. Conclusions & Future Perspectives. Authors didn’t discuss what are the future directions for cardiac MR? How will the results translate to humans?

Author Response

We thank this reviewer for his/her kind and positive words about the quality of our work. Below is our point-by-point response:

1) This is a fair comment raised by this reviewer but we feel that inclusion of such a schematic diagram is beyond the scope of our present review article. Besides, the topic of the MR in cardiac inflammation/fibrosis in general is covered comprehensively in other recent reviews (e.g. see Refs. #1-6 of our manuscript). The goal (and novelty) of our present review is to give an overview of the role of GPCR crosstalk (only) in the MR`s actions in cardiac adverse remodeling.

2) The paper discussed in lines 106-108 is not from our group. Additionally, our manuscript is a literature review that is meant to briefly discuss the main findings of that paper, as well as of all the other original research papers cited in it. Therefore, incorporating any actual data from that paper in our review would be both unethical and beyond the scope of our review article.

3) This is a very valid comment raised by this reviewer and we thank him/her for this. Accordingly, we have now added a couple of sentences in the "Conclusions & Future Perspectives" section of our revised manuscript that address future directions for cardiac MR as they relate to humans, as follows: "At the same time, there is a need for novel and improved MRA agents to successfully  treat heart failure in humans, since the currently used agents are ridden with several adverse effects (e.g. hyperkalemia, renal complications, etc.). Indeed, novel non-steroidal MRA`s (e.g. finerenone), which are purportedly more potent and specific inhibitors at the MR, are currently in development for human heart failure treatment. Given that the MR is expressed throughout the cardiovascular system with various effects in each individual tissue/cell type, it would be advantageous for the pharmaceutic industry to develop novel agents that specifically block the MR only in the myocardium. To this end, targeting the signaling crosstalk mechanisms between the MR and GPCRs that occur specifically in cardiac myocytes could be instrumental"; lines 311-319 (page 7) of the revised manuscript (also highlighted in yellow).

We hope this now satisfies this reviewer.